# EVIDENCE SLOPES AND EFFECTIVE DIMENSION IN SINGULAR LINEAR MODELS

## ABSTRACT

Laplace's approximation and BIC penalize the log evidence by $\frac{d}{2} \log n$ in regular models, where $d$ is the parameter dimension. In singular models, singular learning theory replaces $d/2$ by the real log canonical threshold (RLCT) $\lambda$, an intrinsic exponent that can be strictly smaller. We study linear–Gaussian rank and subspace models where the marginal likelihood is available in closed form and the RLCT is analytically identifiable. We show that the leading Laplace/BIC bias is $(\frac{d}{2} - \lambda) \log n$ and that an RLCT-aware correction recovers the correct evidence slope and removes representation artefacts under overcomplete parametrisations of the same subspace.

## 1 INTRODUCTION

Bayesian model selection compares models via the evidence $Z_n = p(D_n)$. In regular $d$-dimensional models,

$$\log Z_n = \log p(D_n \mid \hat{\theta}_n) - \frac{d}{2} \log n + O(1),$$

which motivates Laplace and BIC. Many modern parametrisations are *singular* (redundant parameters, flat directions): low-rank regression, overcomplete dictionaries, and (in general) neural networks. Singular learning theory predicts instead

$$\log Z_n = \log p(D_n \mid \theta^\star) - \lambda \log n + (m - 1) \log \log n + O(1),$$

where $\lambda$ is the RLCT and $m$ its multiplicity. Replacing $\lambda$ by $d/2$ therefore induces a leading distortion of size $(\frac{d}{2} - \lambda) \log n$.

**Scope.** We choose linear–Gaussian families where (i) the evidence is exactly computable and (ii) RLCT is analytically tractable, yielding a clean ground-truth sandbox to diagnose Laplace/BIC failure mechanisms.

**Contributions.**

- **Rank regression:** for an effective design of rank $r$, the RLCT is $\lambda(r) = r/2$, and the exact evidence is available in closed form.
- **Evidence-slope bias:** Laplace/BIC incurs leading error $(\frac{d}{2} - \lambda(r)) \log n = \frac{d-r}{2} \log n$ in log evidence.
- **Representation invariance:** in a linear dictionary/subspace model, RLCT and leading evidence slope depend on intrinsic subspace dimension $r$ and are invariant under overcomplete reparametrisations, while BIC is not.

## 2 MODELS AND MAIN RESULTS

### 2.1 RANK-$r$ LINEAR–GAUSSIAN REGRESSION

Let $x_i \in \mathbb{R}^p$, $y_i \in \mathbb{R}$, and consider

$$y_i = x_i^\top B\theta + \varepsilon_i, \qquad \varepsilon_i \sim \mathcal{N}(0, \sigma^2),$$

with $B \in \mathbb{R}^{p \times d}$ and Gaussian prior $\theta \sim \mathcal{N}(0, \tau^2 I_d)$. Let $X_n \in \mathbb{R}^{n \times p}$ be the design matrix and $A_n := X_n B \in \mathbb{R}^{n \times d}$ the effective design. If $\mathrm{rank}(B) = r < d$, then $A_n$ has rank at most $r$ and the model is singular in the ambient coordinates.

**Proposition 1** (Exact evidence). *Let $y \in \mathbb{R}^n$ and $A_n \in \mathbb{R}^{n \times d}$ be fixed, with $y \mid \theta \sim \mathcal{N}(A_n \theta, \sigma^2 I_n)$ and $\theta \sim \mathcal{N}(0, \tau^2 I_d)$. Then $Z_n = p(y \mid A_n)$ satisfies*

$$\log Z_n = -\frac{1}{2}\Big(n \log(2\pi) + n \log \sigma^2 + \log \det(I_d + \alpha S_n) + \sigma^{-2}\big(y^\top y - \alpha\, y^\top A_n (I_d + \alpha S_n)^{-1} A_n^\top y\big)\Big),$$

*where $S_n := A_n^\top A_n$ and $\alpha := \tau^2 / \sigma^2$.*

*Proof sketch.* Standard Gaussian integration by completing the square. See Appendix. □

**Proposition 2** (RLCT). *In the rank-$r$ regression setting above, the RLCT is $\lambda(r) = r/2$.*

*Proof sketch.* Only $r$ directions in parameter space curve the likelihood; the remaining $d - r$ are flat. See Appendix. □

**Proposition 3** (Laplace/BIC leading error). *In the rank-$r$ regression setting,*

$$\log Z_n^{\mathrm{Lap}} - \log Z_n = \Big(\frac{d}{2} - \lambda(r)\Big) \log n + O_p(1) = \frac{d - r}{2} \log n + O_p(1),$$

*where $\log Z_n^{\mathrm{Lap}} = \log p(D_n \mid \hat{\theta}_n) - \frac{d}{2} \log n + O_p(1)$.*

*Proof sketch.* Subtract the SLT expansion from the Laplace/BIC expansion and use $\lambda(r) = r/2$. See Appendix. □

**RLCT from evidence slopes (diagnostic).** From $\log Z_n \approx C - \lambda \log n$,

$$\hat{\lambda}_{\mathrm{emp}} := -\widehat{\mathrm{slope}}\big(\log Z_n \text{ vs } \log n\big).$$

(There is *no* extra factor of $1/2$.)

## 2.2 Linear dictionary/subspace model

Let $y_i \in \mathbb{R}^p$ and

$$z_i \sim \mathcal{N}(0, \tau^2 I_d), \qquad y_i = D z_i + \varepsilon_i, \qquad \varepsilon_i \sim \mathcal{N}(0, \sigma^2 I_p),$$

so marginally $y_i \sim \mathcal{N}(0, \Sigma_y(D))$ with $\Sigma_y(D) = \tau^2 D D^\top + \sigma^2 I_p$. The intrinsic dimension is $r = \dim(\mathrm{span}(D)) \leq d$.

**Proposition 4** (Representation invariance). *Let $D \in \mathbb{R}^{p \times d}$ and $D' \in \mathbb{R}^{p \times d'}$ have the same column span of dimension $r$. Then their leading evidence slopes coincide (same $\lambda(r)$), and $\log p(D_n \mid D) = \log p(D_n \mid D') + O_p(1)$ as $n \to \infty$.*

*Proof sketch.* Both parametrisations induce the same rank-$r$ signal covariance family on $y$; the intrinsic exponent depends on the model class, not coordinates. See Appendix. □

**Remark 1** (BIC is not representation-invariant). *A naive BIC score penalizes by $d/2 \log n$ (or $d'/2 \log n$), so it can differ by $\approx \frac{1}{2}(d - d') \log n$ even when $D$ and $D'$ represent the same subspace.*

## 3 Experiments (minimal)

We use synthetic linear–Gaussian data where the exact evidence is computable (Proposition 1). We compare the BIC-style approximation

$$\log Z_n^{\mathrm{BIC}} := \log p(D_n \mid \hat{\theta}_n) - \frac{d}{2} \log n$$

to an RLCT-aware correction

$$\log Z_n^{\mathrm{RLCT}} := \log p(D_n \mid \hat{\theta}_n) - \lambda(r) \log n, \qquad \lambda(r) = r/2,$$

and evaluate errors $\Delta_{\mathrm{BIC}}(n) := \log Z_n^{\mathrm{BIC}} - \log Z_n$ and $\Delta_{\mathrm{RLCT}}(n) := \log Z_n^{\mathrm{RLCT}} - \log Z_n$ as functions of $\log n$.

Figure 1: Rank sweep in linear regression. Estimated slope of $\Delta_{\mathrm{BIC}}(n)$ and $\Delta_{\mathrm{RLCT}}(n)$ versus $\log n$ as a function of intrinsic rank $r$ (ambient $d$ fixed). Consistent with Proposition 3, the BIC slope magnitude increases as $d - r$ grows, while the RLCT-corrected slope remains near zero.

| Quantity | Value |
|---|---|
| Minimal dictionary ($d = r = 3$): $\log Z_{\mathrm{exact}}$ | $-297.53$ |
| Overcomplete dictionary ($d' = 6$): $\log Z_{\mathrm{exact}}$ | $-297.76$ |
| Minimal BIC approximation | $-293.80$ |
| Overcomplete BIC approximation | $-301.75$ |
| Minimal RLCT-aware approximation | $-293.80$ |
| Overcomplete RLCT-aware approximation | $-293.80$ |

Table 1: Minimal vs overcomplete dictionaries spanning the same subspace. Exact evidences differ only by $O(1)$ (Proposition 4), while BIC changes substantially with the number of columns; the RLCT-aware correction is invariant to this reparametrization.

### 3.1 RANK SWEEP (MAIN FIGURE)

Fix ambient dimension $d$ and vary intrinsic rank $r$. For each $r$, compute $\Delta(n)$ on a geometric grid of $n$ and fit a line versus $\log n$.

### 3.2 DICTIONARY REPRESENTATION TEST (MAIN TABLE)

Compare a minimal dictionary $D \in \mathbb{R}^{p \times r}$ to an overcomplete $D' \in \mathbb{R}^{p \times d'}$ with the same span.

## 4 DISCUSSION AND LIMITATIONS

We isolate the leading mechanism behind Laplace/BIC bias in singular models: replacing the intrinsic exponent $\lambda$ by $d/2$ introduces a $(d/2 - \lambda) \log n$ distortion in log evidence. Linear–Gaussian rank/subspace families provide ground truth for this effect because $\log Z_n$ and $\lambda$ are tractable. A limitation is that our experiments are synthetic; extending evidence-slope diagnostics to realistic deep-learning pipelines is a natural next step.

## 5 CONCLUSION

In rank-deficient linear–Gaussian models, Laplace/BIC incurs a leading log-evidence error with slope $(d - r)/2$ in $\log n$, while RLCT-corrected scoring recovers the correct slope. In linear dictionary

models, intrinsic subspace dimension governs evidence/RLCT, whereas BIC depends on the chosen coordinate representation.

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

## A  RELATED WORK

**Laplace approximation and BIC.**  Laplace's method and the Bayesian Information Criterion (BIC) are standard tools for approximating marginal likelihoods in regular parametric models (Tierney & Kadane, 1986; Schwarz, 1978; Kass & Raftery, 1995). Under smoothness and identifiability

assumptions, the posterior concentrates at rate $n^{-1/2}$ around a unique maximum likelihood or MAP estimate $\hat{\theta}_n$, and the log evidence admits the expansion

$$\log p(D_n) = \log p(D_n \mid \hat{\theta}_n) - \frac{d}{2}\log n + O(1), \qquad (1)$$

where $d$ is the parameter dimension. BIC keeps only the data-fit term and the $d/2\log n$ penalty and is widely used for model selection in regression, time series, and latent variable models.

**Singular learning theory.** Many modern models violate the regularity assumptions behind Laplace/BIC: mixture models, neural networks, low-rank and overcomplete representations all exhibit redundant parameters and flat directions in the likelihood. Singular learning theory replaces the dimension $d$ by the RLCT $\lambda$, a birational invariant that quantifies the local singularity of the likelihood at Kullback–Leibler minimisers (Watanabe, 2009; 2018). In such models the evidence behaves as

$$\log p(D_n) = \log p(D_n \mid \theta^\star) - \lambda\log n + (m-1)\log\log n + O(1), \qquad (2)$$

where $\theta^\star$ is any KL minimiser, $\lambda > 0$ is the RLCT, and $m \in \mathbb{N}$ is its multiplicity. For regular models $\lambda = d/2$ and $m = 1$, recovering the classical Laplace/BIC expansion; in singular models typically $\lambda < d/2$. Analytic RLCTs have been derived for certain mixture and rank-constrained models (Watanabe, 2000; 2009; Aoyagi & Watanabe, 2010), but closed-form results are rare beyond simple families.

**Empirical flatness and effective dimension.** Several works propose empirical surrogates for model "flatness" or effective dimension, based on the Hessian spectrum, Fisher information, or PAC-Bayesian bounds (Hochreiter & Schmidhuber, 1997; Keskar et al., 2017; Liang et al., 2019; Dziugaite & Roy, 2017). These proxies often correlate with generalisation in overparameterised networks but are heuristic and not directly grounded in the asymptotic marginal likelihood. Closer to SLT, some recent work attempts to estimate RLCT-like exponents from training curves or posterior samples, typically without access to ground-truth RLCT or exact evidences (Matsuda & Watanabe, 2019; Rissanen, 2007). Our work differs in that we stay within a linear–Gaussian family where RLCT and the exact marginal likelihood are analytically tractable, and we compare these quantities directly to Laplace/BIC.

**Low-rank and overparameterised models.** Low-rank regression, factor analysis, and overcomplete dictionaries are prototypical examples of models that use a low-dimensional subspace but potentially many parameters to represent it (Ghahramani & Hinton, 1998; Tipping & Bishop, 1999; Olshausen & Field, 1997). Model selection in these families is often carried out using BIC-type penalties on the raw number of parameters. From an SLT perspective, however, the effective complexity should depend on the intrinsic rank (the dimension of the subspace) rather than on the number of coordinates used to represent it. The linear–Gaussian models we study provide a setting where this distinction can be made explicit and quantified in terms of marginal likelihood.

## B    PROOFS

### B.1    PROOF OF PROPOSITION 1

*Proof.* By Bayes' rule, the posterior of $\theta$ is Gaussian with precision

$$\Lambda_n = \sigma^{-2}A_n^\top A_n + \tau^{-2}I_d = \sigma^{-2}(S_n + \alpha^{-1}I_d),$$

and mean $\mu_n = \sigma^{-2}\Lambda_n^{-1}A_n^\top y$, where $S_n := A_n^\top A_n$ and $\alpha := \tau^2/\sigma^2$. Writing the joint density $p(y,\theta)$ and integrating over $\theta$ corresponds to completing the square in $\theta$. Standard Gaussian identities yield

$$p(y) = \int p(y \mid \theta)\,\pi(\theta)\,d\theta = (2\pi)^{-n/2}\sigma^{-n}\det(I_d+\alpha S_n)^{-1/2}\exp\Big(-\tfrac{1}{2\sigma^2}\big(y^\top y - \alpha\,y^\top A_n(I_d+\alpha S_n)^{-1}A_n^\top y\big)\Big),$$

and taking logs gives the claimed expression.    $\square$

### B.2   PROOF OF PROPOSITION 2

*Proof.* We outline the argument using the exact evidence expression. Let $S_n = A_n^\top A_n$ and write its spectral decomposition $S_n = U \operatorname{diag}(s_1, \ldots, s_d) U^\top$ with $s_1 \geq \cdots \geq s_d \geq 0$. Under $\operatorname{rank}(B) = r$ and nondegenerate design covariance, $s_1, \ldots, s_r$ scale as $\Theta(n)$ while $s_{r+1}, \ldots, s_d$ are $O(1)$.

Consider the log-determinant term in Proposition 1:

$$\log \det(I_d + \alpha S_n) = \sum_{j=1}^d \log(1 + \alpha s_j) = \sum_{j=1}^r \log(1 + \alpha s_j) + \sum_{j=r+1}^d \log(1 + \alpha s_j).$$

For $j \leq r$, $s_j = \Theta(n)$, so $\log(1 + \alpha s_j) = \log n + O(1)$, contributing

$$\sum_{j=1}^r \log(1 + \alpha s_j) = r \log n + O(1).$$

For $j > r$, $s_j = O(1)$, so $\sum_{j=r+1}^d \log(1 + \alpha s_j) = O(1)$. Since the log evidence contains $-\frac{1}{2} \log \det(I_d + \alpha S_n)$, the leading $\log n$ term is

$$-\frac{1}{2} \log \det(I_d + \alpha S_n) = -\frac{r}{2} \log n + O(1).$$

The remaining quadratic form term in Proposition 1 contributes at most $O(1)$ to the $\log n$ scaling under realizability (it is dominated by the data-fit term plus constants). Thus the leading complexity term in $\log Z_n$ is $-\frac{r}{2} \log n$, so the RLCT is $\lambda(r) = r/2$.  $\square$

### B.3   PROOF OF PROPOSITION 3

*Proof.* From the SLT expansion, in this realizable Gaussian setting,

$$\log Z_n = \log p(D_n \mid \theta^\star) - \lambda(r) \log n + O_p(1).$$

Laplace/BIC yields

$$\log Z_n^{\text{Lap}} = \log p(D_n \mid \hat\theta_n) - \frac{d}{2} \log n + O_p(1).$$

Since $\hat\theta_n \to \theta^\star$ and the data-fit difference $\log p(D_n \mid \hat\theta_n) - \log p(D_n \mid \theta^\star)$ is $O_p(1)$, subtracting gives

$$\log Z_n^{\text{Lap}} - \log Z_n = \left(\frac{d}{2} - \lambda(r)\right) \log n + O_p(1).$$

Using Proposition 2 ($\lambda(r) = r/2$) yields

$$\log Z_n^{\text{Lap}} - \log Z_n = \frac{d - r}{2} \log n + O_p(1).$$

$\square$

### B.4   PROOF OF PROPOSITION 4

*Proof.* In the dictionary model,

$$z_i \sim \mathcal{N}(0, \tau^2 I_d), \qquad y_i = D z_i + \varepsilon_i, \qquad \varepsilon_i \sim \mathcal{N}(0, \sigma^2 I_p),$$

so marginally

$$y_i \sim \mathcal{N}\big(0, \Sigma_y(D)\big), \qquad \Sigma_y(D) = \tau^2 D D^\top + \sigma^2 I_p.$$

If $D$ and $D'$ have the same column span $V \subset \mathbb{R}^p$ with $\dim(V) = r$, then $D D^\top$ and $D' D'^\top$ are positive semidefinite matrices with the same range $V$ and rank $r$. Consequently, both parametrisations induce the same *model class* of rank-$r$ signal-plus-noise covariances on $y$ (up to reparametrisation inside the latent coordinates).

SLT invariance implies that RLCT is a birational invariant of the model class: reparametrising does not change the intrinsic exponent governing the evidence asymptotics. Therefore $\lambda(D) = \lambda(D') = \lambda(r)$.

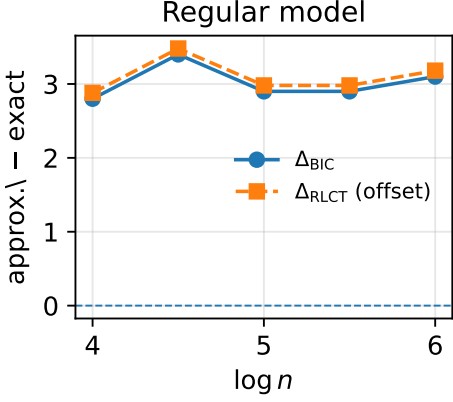

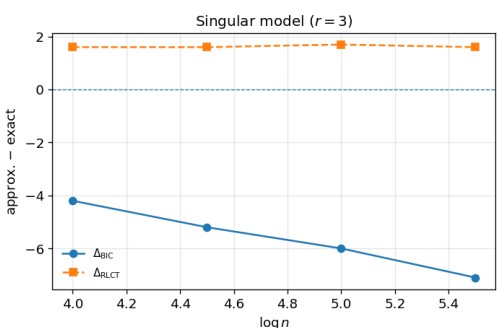

Figure 2: Regular model ($r = d$): $\Delta_{\text{BIC}}(n)$ and $\Delta_{\text{RLCT}}(n)$ versus $\log n$. Both curves are approximately flat (slope near zero), consistent with $\lambda = d/2$.

Figure 3: Singular model ($r < d$): $\Delta_{\text{BIC}}(n)$ and $\Delta_{\text{RLCT}}(n)$ versus $\log n$. The BIC error shows a clear linear trend in $\log n$, while the RLCT-corrected error remains nearly flat.

Moreover, the log evidence in a Gaussian covariance model is a sum of a log-determinant term and a quadratic form term in $\Sigma_y(D)^{-1}$; since both parametrisations correspond to the same rank-$r$ signal family, their large-$n$ leading terms coincide, and any differences (e.g., due to finite-$n$ optimisation or coordinate choices) appear only in $O_p(1)$ constants. Hence

$$\log p(D_n \mid D) = \log p(D_n \mid D') + O_p(1),$$

and in particular the leading $-\lambda(r) \log n$ term is shared. $\qquad\square$

## C  ADDITIONAL EXPERIMENTS/FIGURES (OPTIONAL)

This appendix provides additional visual checks that (i) separate the regular vs. singular regimes, and (ii) illustrate redundancy in overcomplete dictionary parametrisations via eigenspectra.

### C.1  REGULAR VS. SINGULAR ERROR TRENDS

We compare a regular configuration ($r = d$) to a singular configuration ($r < d$) while keeping the remaining generative setup fixed. For each sample size $n$ on a geometric grid, we compute the errors

$$\Delta_{\text{BIC}}(n) := \log Z_n^{\text{BIC}} - \log Z_n, \qquad \Delta_{\text{RLCT}}(n) := \log Z_n^{\text{RLCT}} - \log Z_n,$$

and visualize them against $\log n$.

### C.2  DICTIONARY EIGENSPECTRA AND REDUNDANCY

For the dictionary model, overcomplete parametrisations introduce redundant coordinates. A simple visualization is the eigenspectrum of $D^\top D$ (or equivalently $DD^\top$), which exhibits $r$ nonzero eigenvalues followed by a block near zero when the represented subspace dimension is $r < d$.

### C.3  SLOPE ESTIMATION PROTOCOL (OPTIONAL)

To estimate slopes robustly, we recommend: (i) using a geometric grid of sample sizes $n$, (ii) repeating each $n$ over multiple seeds, and (iii) reporting uncertainty via seed-wise standard deviation or a bootstrap over $(\log n, \Delta(n))$ pairs. This makes explicit when the linear-in-$\log n$ regime is visible and how much data is required to resolve it.

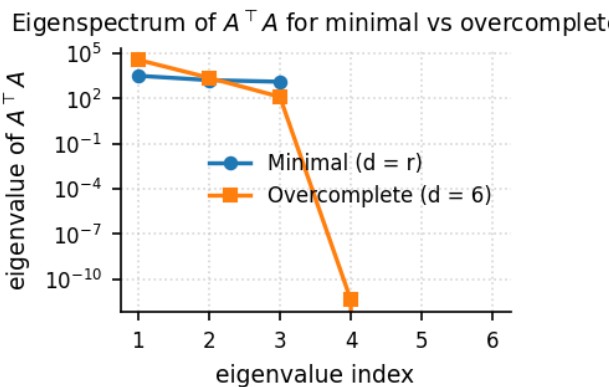

Figure 4: Eigenvalue spectra of $D^\top D$ (minimal dictionary, $d = r$) and $D'^\top D'$ (overcomplete dictionary, $d' > r$) for the same underlying subspace. Both have $r$ large eigenvalues followed by near-zero eigenvalues corresponding to redundant directions.

