# OpenReview forum: "Evidence Slopes and Effective Dimension in Singular Linear Models"
_ICLR.cc/2026/Workshop/Sci4DL — Sci4DL 2026_

### Official Review · Reviewer_6y7i · 2026-02-09

**Fit:** 2
**Significance:** 2
**Confidence:** 3

**Summary:**

In model selection, Laplace/BIC is typically used as an approximation to a model’s generalization performance. However, the Laplace approximation fails for low-rank and/or overparameterized models. In this work, the authors derive a better generalization measure based on singular learning theory in linear–Gaussian rank regression, which allows them to analytically understand why BIC does not work in this case (compared to a better generalization measure).

**Strengths:**

The authors are attempting to address a very important question: how deep learning (overparameterized) models generalize in the first place. They provided very clean analytical results, which are validated by simulations (in a simplified regression setting).

**Suggestions:**

While I really like the overall conceptual idea and the direction the authors are heading, the results here feel too preliminary and a bit far from the theme of this workshop (i.e., to understand deep learning). It is already well known that the Laplace/BIC approximation does not work for deep learning models. While I lack the context to judge how novel the analytical results are for the RLCT in the regression setting, the current results still feel far from deep learning models.

To really demonstrate the value of the theory, I think the authors either need to show, even in this simple setting, a phenomenon that is relevant to understanding deep learning models (e.g., there are work showing double descent and benign overfitting in linear models, can the RLCT be used to understand these in a precise and quantitative way?); or alternatively, attempt to apply the theory to more DL models, maybe numerically calculate some version of the RLCT bound.

---

### Official Review · Reviewer_5hQj · 2026-02-18

**Fit:** 2
**Significance:** 2
**Confidence:** 2

**Summary:**

This paper investigates the discrepancy between classical marginal likelihood approximations (Laplace and BIC) and the exact log evidence in a rank-deficient linear-Gaussian regression model. They show that an RLCT-aware correction correctly recovers the evidence slope while Laplace and BIC are biased.

**Strengths:**

The authors propose a simple linear model to demonstrate the limitation of Laplace/BIC when dealing with rank-deficient models. This serves as an excellent minimal model for understanding how overparameterization distorts standard model selection criteria.

**Suggestions:**

While the linear model serves as a minimal analytical model to understand the failure of Laplace/BIC, the authors should clarify whether their analytical results are new derivations or applications of known results (e.g., Watanabe, 2000; Aoyagi & Watanabe, 2010) and how generalizable this linear model is to more complex singular models.

---

### Official Review · Reviewer_nrnw · 2026-02-24

**Fit:** 2
**Significance:** 2
**Confidence:** 2

**Summary:**

The paper shows that the log of the marginal likelihood (evidence) of singular models–ones with flat directions in parameter space–have a correction term corresponding with $\lambda$, the 'Real Log Canonical Threshold' (RLCT). The RLCT serves as an intrinsic, variable exponent of the likelihood. After accounting for the correction term, the correct evidence can be recovered for overparameterized models.

**Strengths:**

The paper gives analytic solutions to the evidence of the linear-Gaussian and dictionary models. Given RLCT corrections are shown empirically to correctly account for varying degrees of parameter counts of dictionary learning. The dictionary learning corrections appear to be useful to the area of Mechanistic Interpretability.

**Suggestions:**

As someone who is not innately familiar with singular learning theory, the concepts and presentation of the main content was lacked motivation, especially without relevant citations (eg line ~27 has the unsubstantiated claim "Many modern parametrisations are singular, ..., (in general) neural networks," are these trained neural networks? In which regimes and on what tasks? I'm not convinced this is true.) The also paper appears to lack a proper introduction, as the first paragraphs appear to discuss setup rather than concrete motivation, which may contribute to the lack of clarity when reading the paper. As a general recommendation, I would highlight a greater need for discussion rather than equations--the story here isn't clear, even with the derivations (appearing on inspection to be) correct. Experimental details, particularly those corresponding with Figure 1 & Table 1 visually detailing the main results, are sparse, and only focus on low-dimensional data without proper justification.

---

### Meta-Review · Area_Chair_8u8a · 2026-02-28

**Recommendation:** Accept

**Metareview:**

Reviewers were generally positive, so I'll accept this. My main comment's that this workshop is on the *science* of DL, not the mathematics. As reviewers also asked, what's the scientific contribution here? How does this connect to empirics?

Another major comment's that this is really not written well for people who aren't already in the SLT community. Surely you should write out "Bayesian info criterion" before using the acronym, and certainly you should define it and motivate it, right?

---

### Decision · Program_Chairs · 2026-03-02

Accept